# Adaptive Skip Intervals: Temporal Abstraction for Recurrent Dynamical Models

**Alexander Neitz**[1][3]    **Giambattista Parascandolo**[1][2]    **Stefan Bauer**[1][2]    **Bernhard Schölkopf**[1][2]

[1]Max Planck Institute for Intelligent Systems
[2]Max Planck ETH Center for Learning Systems
[3]`aneitz@tue.mpg.de`

## Abstract

We introduce a method which enables a recurrent dynamics model to be temporally abstract. Our approach, which we call Adaptive Skip Intervals (ASI), is based on the observation that in many sequential prediction tasks, the exact time at which events occur is irrelevant to the underlying objective. Moreover, in many situations, there exist prediction intervals which result in particularly easy-to-predict transitions. We show that there are prediction tasks for which we gain both computational efficiency and prediction accuracy by allowing the model to make predictions at a sampling rate which it can choose itself.

## 1    Introduction

A core component of intelligent agents is the ability to predict certain properties of future states of their environments (Legg and Hutter, 2007). For example, model-based reinforcement learning (Daw, 2012; Arulkumaran et al., 2017) decomposes the task into the two components of learning a model and then using the learned model for planning ahead.

Despite significant recent advances, even relatively simple tasks like pushing objects is still a challenging robotic task and foresight for robot planning is still limited to relatively short horizon tasks (Finn and Levine, 2017). This is partially due to the fact that errors even from early stages in the prediction pipeline are accumulating especially when new or complex environments are considered.

Many dynamical systems have the property that long-term predictions of future states are easiest to learn if they are obtained by a sequence of incremental predictions. Our starting point is the hypothesis that at each instant of the evolution, there is an ideal *temporal step length* associated with those state transitions which are easiest to predict: Intervals which are too long correspond to complicated mechanisms that could be simplified by breaking them down into a successive application of simpler mechanisms. On the other hand, intervals which are too short do not contain much change, which means that the predictor has to represent roughly the identity – this can lead to a situation where the model makes small absolute errors $\delta s$, but a large relative error $\frac{\delta s}{\Delta t}$, which is the rate at which the prediction error accumulates. This tradeoff is illustrated in Figure 1. An additional drawback of too short prediction intervals is that it requires many predictions, which can be computationally expensive. Somewhere in-between the two extremes, there is an ideal step length corresponding to transitions that are easiest to represent and learn.

We propose Adaptive Skip Intervals (ASI), a simple change to autoregressive environment simulators (Chiappa et al., 2017; Buesing et al., 2018) which can be applied to systems in which it is not necessary to predict the exact time of events. While in the literature, abstractions are often considered with respect to hierarchical components e.g. for locomotor control (Heess et al., 2016) or expanding the dynamics in a latent space (Watter et al., 2015), our work focuses on temporal abstractions. Our goal is to understand the dynamics of the environment in terms of robust causal mechanisms at the

right level of temporal granularity. This idea is closely related to causal inference (Peters et al., 2017) and the identification of invariances (Pearl, 2009; Schölkopf et al., 2012; Peters et al., 2016) and mechanisms (Parascandolo et al., 2017).

ASI allows the model to dynamically adjust the temporal resolution at which predictions are made, based on the specific observed input. In other words, the model has the option to converge to the easiest-to-predict transitions, with prediction intervals $\Delta t$ that are not constant over the whole trajectory, but situation-dependent. Moreover, the model is more robust to certain shifts in the evolution speed at training time, and also to shifts to datasets where the trajectories are partly corrupted. For example, when some frames are missing or extremely noisy, a frame-by-frame prediction method would be forced to model the noise, especially if it is not independent of the state. Flexibly adjusting the time resolution of predictions also results in more computationally efficiency, as fewer steps need to be predicted where they are not necessary — a key requirement for real-time applications.

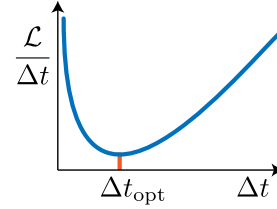

Figure 1: Hypothesized relationship between skip interval $\Delta t$ and error accumulation rate $\frac{\mathcal{L}}{\Delta t}$.

A type of prediction task which can especially profit from our proposed method is one which exhibits a property we call *inconsequential chaos*. To illustrate this, consider the following example: In Figure 2 we visualize the trajectories of a ball which falls into a funnel-shaped object at different initial horizontal velocities. The exact trajectories that are taken within the funnel depend sensitively on the initial state and are therefore difficult to predict ahead of time. On the other hand, predicting that the ball will hit the horizontal platform on the bottom is easy because it only requires knowing that when the ball falls somewhere into the funnel, it will come out at the bottom end, irrespective of how long it bounces around. If we are only interested in predicting where the ball will ultimately land, we can skip the difficult parts, provided that they are inconsequential. Figure 3 explains another perspective to motivate our method.

## 2 Preliminaries

### 2.1 Problem statement

The machine learning problem we are considering is a classification problem where the labels are generated by a dynamical process, such as a Hidden Markov Model. As auxiliary data, we get access to observations of the system's internal state. The training data consists of observation sequences $\{x^{(i)}\}_{i\in 1,\dots,N}$ and labels $\{y^{(i)}\}_{i\in 1,\dots,N}$. The trajectories $x$ are ordered sequences of elements $x_t$ from an observation space $\mathcal{X}$. Typically, a trajectory $x$ arises from repeatedly measuring the dynamical system's state at some fixed sampling rate. To keep the scope limited, we assume the labels $y^{(i)}$ to be categorical, i.e. belonging to a finite set $\mathcal{Y}$. In our formulation, there is only a single label for each trajectory, which intuitively corresponds to the eventual "outcome" of the particular system evolution. At test time, we are only given some initial observations $(x_0, x_1, \dots, x_k)$, for some small $k$ (e.g., $k = 0$ in the fully-observable case) and have to predict the corresponding label $y$.

Note that the problem does not demand the prediction of any future observations $x_t$. As a performance measure we use the accuracy of the label predictions. The role of the classification task is to provide

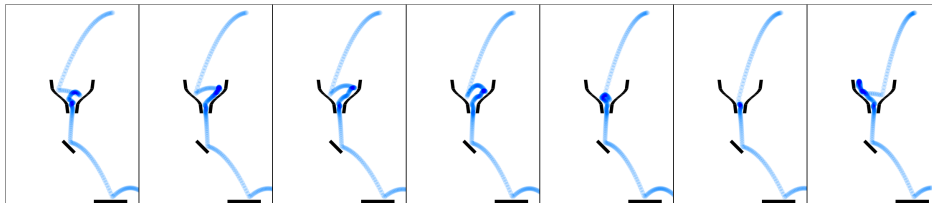

Figure 2: Visualization of a ball which is dropped into a funnel at different initial horizontal velocities. The part of the trajectory within the funnel can be considered *inconsequential chaos*.

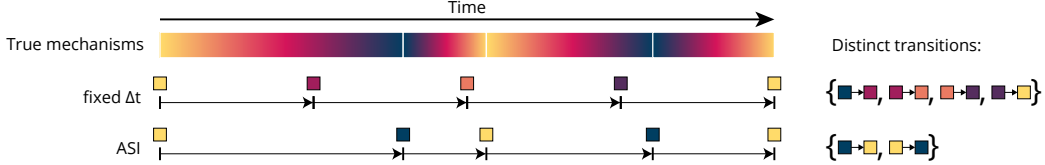

Figure 3: One way to motivate the need for adaptive skip intervals compared to a fixed temporal coarsening is to consider the complexity of the learned model. If the underlying true dynamics have recurring "mechanisms" which take different amounts of time, ASI enables the model to represent fewer distinct transition types, reducing the required model capacity and thus the amount of training data.

a way to measure performance, as the objective is to know how well the model is suited to predict the *qualitative* outcome of each instance. We explicitly do not care about the loss in pixel space. Since frames may be skipped, video-prediction metrics are not relevant for this task. In the future we would like to use our model in latent spaces as well.

It is straightforward to generalize the classification task to a value prediction task in a (hierarchical) reinforcement learning setting, given a fixed policy (e.g. an option, as introduced in Sutton et al. (1999)). However, in this work we focus on uncontrolled tasks only.

## 2.2 Environment simulators

Environment simulators are models which approximate the conditional probability distribution

$$P(X_{t+1}, R_{t+1}|X_t) \tag{1}$$

where $X_t$ is a random variable with range $\mathcal{X}$ which describes the Markovian state of the system at time $t$. $R_t$ is the random variable over some real-valued *cumulant* which we want to track for our task. In order to simplify our experiments, in this paper we consider the special case of *fully-observable* tasks. For this reason, we use the terms "observation" and "state" interchangeably. However, note that in realistic applications, it may be desirable to predict future states given past observations, which poses the additional challenge of state inference. As an additional simplification, we consider deterministic simulators, which put a probability point mass of one on a single future state. For a recent, more detailed investigation of several efficient state-space architectures, see Buesing et al. (2018).

Note that given a distribution over an initial $X_0$, we can apply an environment simulator multiple times to a distribution over the initial state, yielding a probability distribution over trajectories and cumulants.

$$P(X_{0:N}, G_{0:N}) = P(X_0) \prod_{t=1}^{N} P(X_t, G_t|X_{t-1}) \tag{2}$$

*Temporally abstract environment simulators* only need to represent a relaxed version of the above conditional probability distribution:

$$P(X_{t+\tau}, R_t^\tau|X_t) \tag{3}$$

where $\tau$ is some arbitrary time skip interval up to the end of the trajectory, which can be chosen by the model and $R_t^\tau$ denotes the sum $\sum_{k=t}^{\tau} R_k$. In other words, a temporally abstract environment simulator must only be able to predict *some* future state of the system and additionally provide the sum of the cumulants since the last step. To address the classification problem defined in Section 2.1, we only consider tasks where the cumulant is zero everywhere except for the last state of the trajectory, which is a plausible restriction if the cumulant tracks some form of "outcome" of the trajectory.

The dynamical models we consider in this paper consist at their core of a deep neural network $f : \mathcal{X} \to \mathcal{X}$ which is meant to represent the dynamical law of the environment. In order to learn to predict multiple time-steps into the future, $f$ is iterated multiple times, which makes the architecture a recurrent neural network. As the model predicts the new state at time $t + 1$, it needs to be conditioned

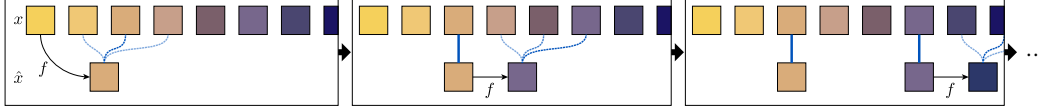

Figure 4: Visualization of the first three steps of ASI with a horizon of $H = 3$. The blue lines represent loss components between the ground truth frames $x$ and predicted frames $\hat{x}$. For simplicity, we do not consider scheduled sampling here, therefore $f$ is always applied to the previous predicted state.

on the previous state at the previous time step $t$. During training, there is a choice for the source of the next input frame for the model: Either the ground truth (observed) frame or the model's own previous prediction can be taken. The former provides more signal when $f$ is weak, while the latter matches more accurately the conditions during inference, when the ground truth is not known. We found the technique of scheduled sampling (Bengio et al., 2015) to be a simple and effective curriculum to address the trade-off described above. Note that other works, such as Chiappa et al. (2017) and (Oh et al., 2017) have addressed the issue in different ways. The exact way of dealing with this issue is orthogonal to the use of temporal abstraction.

## 3 Adaptive skip intervals for recurrent dynamical models

We now introduce a method to inject temporal abstraction into deterministic recurrent environment simulators.

**Training process** The main idea of ASI is that the dynamical model $f$ is not forced to predict every single time step in the sequence. Instead, it has the freedom to skip an arbitrary number of frames up to some pre-defined horizon $H \in \mathbb{N}$. We train $f$ in such a way that it has the incentive to focus on representing those transitions which allow it to predict extended sequences which are accurate over many time steps into the future. Figure 4 visualizes the three steps of the ASI training procedure with a horizon of $H = 3$.

At training time, we feed the first frame $x_1$ into a differentiable model $f$, producing the output $\hat{x}_1 := f(x_1)$. In contrast to classical autoregressive modeling, $\hat{x}_1$ does not have to correspond to the next frame in the ground truth sequence, $x_2$, but can be matched with any frame from $x_2$ to $x_{2+H}$. Importantly, $f$ is not required to know how many frames it is going to skip – the temporal matching is performed by a "training supervisor" who takes $f$'s prediction and selects the best-fitting ground-truth frame to compute the loss, which is later on reduced using gradient based optimization.

To soften the winner-takes-all mechanism, we use an *exploration-curriculum*. At every step, a Bernoulli trial with probability $\mu$ decides whether an exploration or an exploitation step is executed: In an exploration step, the supervisor selects a future frame at random with a frame-skip value between 1 and $H$; in an exploitation step, the supervisor takes the best-fitting ground-truth frame $x_i = \arg\min_{t \in \{2..2+H\}} \mathcal{L}_x(\hat{x}_b, x_t)$ to provide the training signal. At the beginning of training, $\mu$ is high, such that exploration is

---

**Algorithm 1:** Dynamical model learning with ASI

**Input :** $i$'th trajectory $\mathbf{x}^{(i)} = (x_1, x_2, ..., x_{T_i}) \in \mathcal{X}^{T_i}$
Differentiable model $f : \mathcal{X} \to \mathcal{X}$ w/ params $\theta$
Loss function $\mathcal{L} : \mathcal{X} \times \mathcal{X} \to \mathbb{R}$
Matching-horizon $H \in \mathbb{N}$
Exploration schedule $\mu : \mathbb{N} \to [0, 1]$
Scheduled sampling temperatures $\epsilon : \mathbb{N} \to [0, 1]$

$t \leftarrow 1, u \leftarrow 1$  ▷ *Data timestep $t$, abstract timestep $u$*
$l \leftarrow 0$  ▷ *Trajectory loss*
$p \leftarrow x_1$  ▷ *Next input to the dynamics model $f$*
**while** $t < |x|$ **do**
$\quad \hat{x}_u \leftarrow f(p)$
$\quad T \leftarrow \min(t + H, |x|)$  ▷ *Upper time step limit*
$\quad$ **if** Bernoulli$(\mu(i)) = 0$ **then**
$\quad\quad t \leftarrow \arg\min_{t' \in \{t+1..T\}} \mathcal{L}(x_u, x_{t'})$
$\quad$ **else**
$\quad\quad t \sim \text{unif}\{t + 1, T\}$  ▷ *Exploration*
$\quad$ **end**
$\quad l \leftarrow l + \mathcal{L}(x_u, x_t)$  ▷ *Accumulate trajectory loss*

$\quad p \leftarrow \text{binary\_choice}(\hat{x}_u, x_t; p = \epsilon(i))$
$\quad$ ▷ *Scheduled sampling (Bengio et al., 2015)*
$\quad u \leftarrow u + 1$
**end**
$\theta \leftarrow$ gradient descent step on $\theta$ to reduce $l$

---

encouraged. Over the course of several epochs, $\mu$ is gradually decreased such that $f$ can converge to predicting sharp mechanisms. The goal of the exploration schedule is to avoid being caught in a local optimum early on during training. Over the course of the learning process, we gradually decrease the

chance of picking a random frame, effectively transitioning to the winner-takes-all mechanism. We refer to this curriculum scheme *Exploration of temporal matching*.

The best fitting frame $x_i$ is then fed into $f$ again, iterating the same procedure as described above, but from a later starting point. At every step, we accumulate a loss $l_x$, leading to an overall *prediction loss $\mathcal{L}_x$* which is simply the mean of all the step-losses. We train the model $f$ via gradient descent to reduce the prediction loss $\mathcal{L}_x$.

In the example with the funnel, this could intuitively work as follows: the transition from the ball which falls into the funnel to the ball which is at the end of the funnel is the most robust one (let's call it the "robust transition") – it occurs virtually every time. All other positions within the funnel are visited less often. Therefore, $f$ will tend to get most training signal from the robust transition. Hence, $f$ will begin to predict something that resembles the robust transition, which will subsequently be reinforced because it will often be the best-fitting transition which wins in the matching process.

Instead of using a greedy matching algorithm it is conceivable to use a global optimization method which is applied to the whole sequence of iteratively predicted frames, which would then be aligned in the globally best possible way to the ground truth data. However, in this case, we would not be able to alternate randomly between the input sources for $f$, as we currently do with scheduled sampling, because in order to know which ground truth frame to take next, we already need to know the alignment.

Besides exploration of temporal matching,as mentioned in Section 2.2 we adopt another curriculum scheme, *scheduled sampling* (Bengio et al., 2015), which gradually shifts the training distribution from observation-dependent transitions towards prediction-dependent transitions.

**Predicting the labels**   Since the learning procedure can choose to skip difficult-to-predict frames, the mean loss in pixel space would not be a fair metric to evaluate whether ASI serves a purpose. As explained in Section 2.1, one of our central assumptions is that we are dealing with environments which have the notion of a qualitative outcome, represented e.g. by the classification problem associated with the task. Therefore, as a way to measure the learning success, we let a separate classifier $\psi : \mathcal{X} \to \mathcal{P}(\mathcal{Y})$ predict the label of the underlying classification task based on the frames predicted by $f$. At test time, $f$ can unfold the dynamics over multiple steps and $\psi$ is applied to the resulting frames, allowing the combined model to predict the label from the initial frame.

In principle, the classifier $\psi$ could be trained alongside the model $f$, or after convergence of $f$ – the two training processes do not interfere with each other. For the experiments described in Section 4, we hand-specify a classifier $\psi$ ahead of time for each environment. Since our classification tasks are easy, given the last frame of a trajectory, the classifiers are simple functions which achieve perfect accuracy when fed the ground truth frames.

# 4   Experiments

We demonstrate the efficacy of our approach by introducing two environments for which our approach is expected to perform well. Code to reproduce our experiments is available at `https://github.com/neitzal/adaptive-skip-intervals`.

## 4.1   Domains

**Room runner**   In the Room runner task, an agent, represented by a green dot, moves through a randomly generated map of rooms, which are observed in 2D from above. The agent follows the policy of always trying to move towards and into the next room, until it reaches a dead end. Two rooms are colored – the actual dead end which the agent will reach and another room, which is a dead end for another path. One of these two rooms is red, the other one blue, but the assignment is chosen by a fair coin flip. The underlying classification task is to predict whether the agent will end up in the red room or in the blue one. Since there is always exactly one passage between two adjacent rooms, the final room is always well-defined and there is no ambiguity in the outcome. We add noise to the runner's acceleration at every step, simulating an imperfect controller – for example one which is still taking exploratory actions in order to improve.

Figure 5 shows examples for the first states and the resulting trajectories.

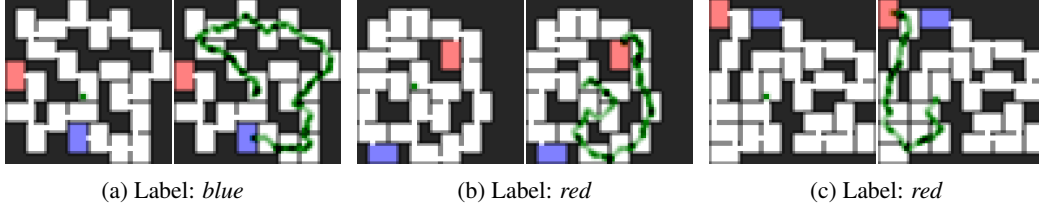

|                    |                   |                  |
|:------------------:|:-----------------:|:----------------:|
| (a) Label: *blue*  | (b) Label: *red*  | (c) Label: *red* |

Figure 5: Examples of first states of the *Room runner* domain, along with the corresponding trajectories which arise from evolving the environment dynamics and the agent's policy. Darker regions in the trajectory correspond to parts where the agent was moving more slowly.

**Funnel board** In this task, a ball falls through a grid of obstacles onto one of five platforms. Every other row of obstacles consists of funnel-shaped objects, which are meant to capture the ball and release it at a well-defined exit position. Variety arises from the random rotations of the sliders, from the random presence or absence of funnels in every layer except for the last one, and from slight perturbations in the funnel and slider positions. The courses are generated such that the ball is always guaranteed to hit exactly one of the platforms. Figure 6 shows three examples for the first states and the ball's resulting paths. In order to simplify the problem, we make the states nearly fully observable by preprocessing the video frames such that they include a trace of the ball's position at the previous step.

The underlying classification task is to predict, given only access to the first frame, on which of the five platforms the ball will land eventually. Note that the task does not include predicting the time when the ball will reach its goal.

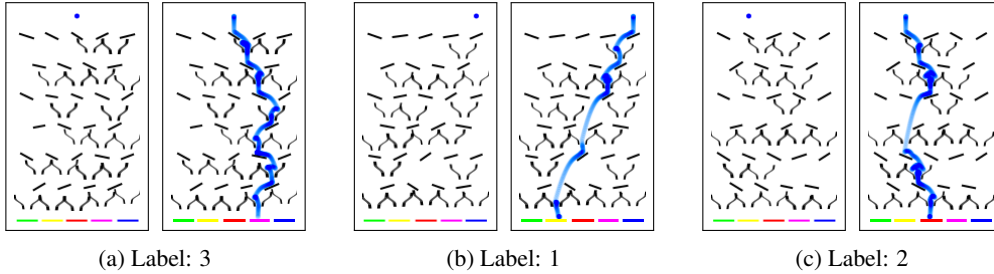

|                  |                  |                  |
|:----------------:|:----------------:|:----------------:|
| (a) Label: 3     | (b) Label: 1     | (c) Label: 2     |

Figure 6: Examples of first states of the *Funnel board* domain, along with the corresponding trajectories which arise from evolving the environment dynamics. The trajectories are merged into one image for visualization purposes only – in the dataset every frame is separate.

## 4.2 Experiment setup

The experiments are ablation studies of our method. We would like to investigate the efficacy of adaptive skip intervals and whether the exploration schedule is beneficial to obtain good results. For each of our two environments, we compare four methods: (a) The recurrent dynamics model with adaptive skip intervals as described in Section 3. (*ASI*) (b) The dynamics model with adaptive skip intervals, but without any exploration phase, i.e. $\mu = 0$. (*ASI w/o exploration*) (c) The dynamics model *without* adaptive skip intervals such that it is forced to predict every step (*fixed ($\Delta t = 1$)*). (d) The dynamics model without adaptive skip intervals such that it is forced to predict every *second* step (*fixed ($\Delta t = 2$)*). In each experiment we train with a training set of 500 trajectories, and we report validation metrics evaluated on a validation set of 500 trajectories. We perform validation steps four times per epoch in order to obtain a higher resolution in the training curves.

For our experiments, we use a neural network with seven convolutional layers as the dynamics model $f$. Architectural details, which are the same in all experiments, are described in the Appendix. Like (Weber et al., 2017), we train $f$ using a pixel-wise binary cross entropy loss. Hyperpararameter settings such as the learning rates are determined for each method individually by using the set of parameters which led to the best result (highest maximum achieved accuracy on the validation set), out of 9 runs each. We use the same search ranges for all experiments and methods. The remaining

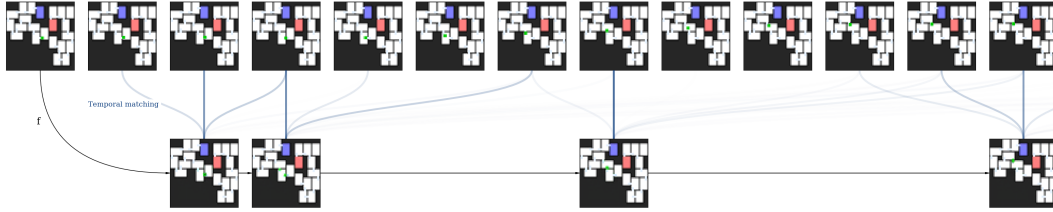

Figure 7: Portion of a sequence from Room runner using ASI, with ground truth frames on top and predicted, temporally aligned sequence on bottom.

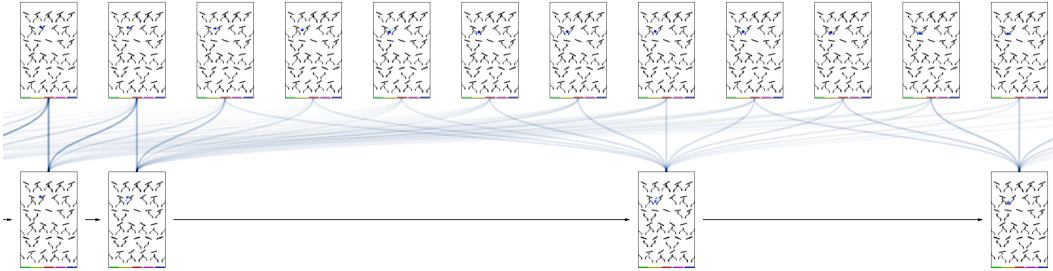

Figure 8: Portion of a sequence from Funnel board using ASI, with ground truth frames on top and predicted, temporally aligned sequence on bottom. Darker lines connecting a predicted frame to the ground truth frames correspond to better matching in terms of pixel loss.

hyperparameters, including search ranges, are provided in the Appendix. For instance, as a value for the horizon $H$ in the ASI runs, our search yielded optimal results for values of around 20 in both experiments. After fixing the best hyperparameters, each method is evaluated 8 additional times with different random seeds, which we use to report the results. We additionally included baselines with $\Delta t > 2$, but to reduce the amount of computation did not perform another hyperparameter search for them, instead taking the best parameters for the baseline "fixed ($\Delta t = 2$)".

## 4.3 Results

We begin by visualizing how the network with adaptive skip intervals performs after training. In Figure 8 we show a portion of one trajectory from the Funnel board, as processed by the network. As shown, the network trained with ASI has learned to skip a variable number of frames, specifically avoiding the bouncing in the funnel, and directly predicting the exiting ball. Similarly, Figure 7 shows a portion of a sequence from the Room runner domain. As the videos presented at `http://tiny.cc/x2suwy` demonstrate, ASI is able to produce sharp predictions over many time-steps while the fixed-skip baselines produce blurry predictions.

**Quantitative results**  As shown in Figure 9, ASI outperforms the fixed-steps baselines on both datasets. On *Funnel board* the networks equipped with adaptive skip intervals achieve higher accuracy *and* in a shorter time, with exploration of adaptive skip intervals obtaining even better results. In the

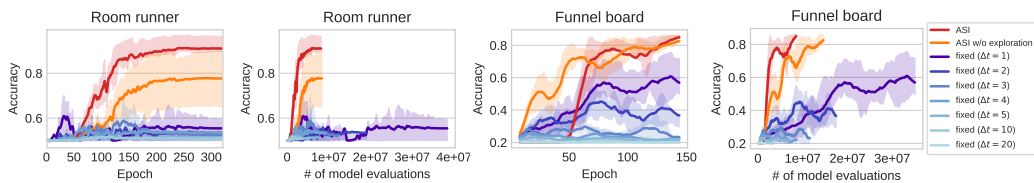

Figure 9: Learning progress, curves show validation accuracies on two tasks. For each task, we show on the horizontal axis the number of model evaluations and the epoch number. Curves show mean validation accuracy, evaluated on 500 trajectories. The training sets consist of 500 trajectories in each experiment. Shaded areas correspond to the interquartile range over all eight runs.

*Room runner* task, we observe a significant improvement of ASI with exploration over the version without exploration and the baselines. Note that some of the baselines curves get worse after an initial improvement. This can be explained by the fact that the two training curricula, scheduled sampling and exploration of temporal matching, create a nonstationary distribution for the network. We observe that ASI appears more resilient to this effect.

**Computational efficiency**    Note that the x-axis in Figure 9 represents the number of forward-passes through $f$, which loosely corresponds to the wall clock time during the training process. Since the adaptive skip intervals methods are allowed to skip frames, they need fewer model evaluations (and therefore fewer backpropagation passes at training time) than fixed-rate training schemes. In the tasks we considered, not only this gain in training speed does not come at the cost of reduced accuracy, but it actually improves the overall performance. Full-resolution timelines can be viewed at `http://tiny.cc/x2suwy`

**Robustness w.r.t. perturbation of dynamics**    Another advantage of the temporally abstract model which we hypothesize is that the training process is more stable when the dynamical systems changes in a certain way. This is relevant because in real systems, the i.i.d. assumption is often violated. The same is true for reinforcement learning tasks, in which the distribution over observed transition changes as the agent improves its policy or due to changes in the environment over time. As a test for our hypothesis, we prepare a second version of the Funnel board dataset with 500 trajectories of slightly altered physics: The bounciness of the funnel walls is reduced to zero. This leads to a slightly different behavior in the funnels, but the final platforms are the same in the majority of trajectories. We start with the perturbed version and before the start of the 75th epoch, we exchange it with the original one. Figure 10 shows the accuracy curves for this experiment. We observe that while the fixed frame-rate baselines learn the correct classification better than in the more difficult original task, after the switch the validation accuracy quickly deteriorates. Note that freezing the network at epoch 75 would leave the validation accuracy almost unchanged, since both versions of the task have similar labels.

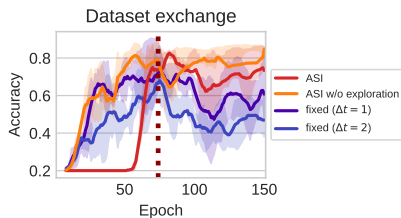

Figure 10: Up to epoch 75 we use a version of the Funnel board task where the funnels' bounciness is set to zero. At epoch 75 we switch the dataset for the standard one but otherwise keep the training procedure going.

## 5    Related work

The observation that every environment has an optimal sampling frequency has also been made for reinforcement learning. For instance, Braylan et al. (2000) investigate the effect of different frame-skip intervals on the performance of agents learning to play Atari 2600 games. A constant frame-skip value of four frames is considered standard for Deep RL agents (Machado et al., 2017). Focusing on spatio-temporal prediction problems, (Oh et al., 2015) introduce a neural network architecture for action conditional video prediction. Their approach benefits from using curriculum learning to stabilize the training of the network. Buesing et al. (2018) investigate action-conditional state-space models and explicitly consider "jumpy" models which skip a certain number of timesteps in order to be more computationally efficient. In contrast to our work they do not use adaptive skip intervals, but skip at a fixed frame rate. Belzner (2016) introduces a time-adaptive version of model-based online planning in which the planner can optimize the step-length adaptively. Their approach focuses on temporal abstraction in the space of actions and plans. Temporal abstraction in the planning space is also a motivation of the field of hierarchical reinforcement learning (Barto and Mahadevan, 2003), often in the framework of semi-MDPs – Markov Decision Processes with temporally extended actions (e.g. Puterman, 1994).

The idea of skipping time steps has also been investigated in Ke et al. (2017), where the authors present a way to attack the problem of long-term credit assignment in recurrent neural networks by only propagating errors through selected states instead of every single past timestep.

Closely related to our work is the Predictron (Silver et al., 2016), which is a deep neural network architecture which is set up to perform a sequence of temporally abstract lookahead steps in a latent space. It can be trained end-to-end in order to approximate the values in a Markov Reward Process. In contrast to ASI, the outputs of the Predictron are regressed exclusively towards rewards and values, which circumvents the need for an explicit solution to the temporal alignment problem. However, by ignoring future states, the training process ignores a large amount of dynamical information from the underlying system.

Similar in spirit to the Predictron, the value prediction network (VPN) (Oh et al., 2017) proposes a neural network architecture to learn a dynamics model whose abstract states make option-conditional predictions of future values rather than of future observations. Their temporal abstraction is "grounded" by using option-termination as the skip-interval.

Ebert et al. (2017) introduced temporal skip connections for self-supervised visual planning to keep track of objects through occlusion.

(Pong et al., 2018) introduce temporal difference models (TDM) which are dynamical models trained by temporal difference learning. Their approach starts with a temporally fine-grained dynamics model, which is represented with a goal-conditioned value function. The temporal resolution is successively coarsened so as to converge toward a model-free formulation.

Concurrently to our work, Jayaraman et al. (2018) propose a training framework with a similar motivation to ours. They further explore ways to generalize the objective and include experiments on hierarchical planning.

## 6   Conclusion

We presented a time skipping framework for the problem of sequential predictions. Our approach builds on concepts from causal discovery (Peters et al., 2017; Parascandolo et al., 2017) and can be included in multiple fields where planning is important. In cases where our approach fails, e.g. when the alignment of predicted and ground truth is lost and the model does not have the power to restore it, more advanced optimization methods like dynamic time warping (Müller, 2007) during the matching phase may help at the cost of the simplicity and seamless integration of the scheduled sampling, as described in Section 3.

An interesting direction for future work is the combination of temporal abstraction with abstractions in a latent space. As noted for instance by Oh et al. (2017), predicting future observations is a too difficult task for realistic environment due to the high dimensionality of typical observation spaces.

The idea of an optimal prediction skip interval should extend to the case of stochastic generative models, where instead of a deterministic mapping from current to next state, the model provides a probability distribution over next states. In this case, ASI should lead to simpler distributions, allowing for simpler models and more data efficiency just as in the deterministic case. The evaluation of this claim is left for future work.

Another line of investigation which is left to future work is to integrate ASI with action-conditional models. As mentioned in Section 2.1, the problem could be addressed by using a separate ASI-dynamical model for each policy or option, which would allow for option-conditional planning. However, there may be a more interesting interplay between ideal skip intervals and switching points for options, which suggest that they should ideally be learned jointly.

**Acknowledgements**

This work is partially supported by the International Max Planck Research School for Intelligent Systems and the Max Planck ETH Center for Learning Systems.

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
