[Supplementary Material]

# SUPPLEMENTARY MATERIAL

## 6.1 Full algorithm with comments

---

**Algorithm 2:** Dynamical model learning with ASI

---

**Input :** Dataset of $N$ trajectories $\{(x_1, x_2, ..., x_{T_i})\}_{i=1}^{N}$; each $x_t \in \mathcal{X}$
    Differentiable model $f : \mathcal{X} \to \mathcal{X}$ with parameters $\theta$
    Loss function $\mathcal{L} : \mathcal{X} \times \mathcal{X} \to \mathbb{R}$
    Matching-horizon $H \in \mathbb{N}$
    Exploration schedule $\mu : \mathbb{N} \to [0, 1]$
    Scheduled sampling temperatures $\epsilon : \mathbb{N} \to [0, 1]$
$\theta \leftarrow$ Initialize model parameters
training_step $\leftarrow 0$
**repeat**
    $x \leftarrow$ get next trajectory from dataset
    $t \leftarrow 1, u \leftarrow 1$     ▷ *Ground truth timestep $t$ and abstract timestep $u$*
    $l \leftarrow 0$     ▷ *Trajectory loss*
    $p \leftarrow x_1$     ▷ *Next input to the dynamics model $f$*
    **while** $t < |x|$ **do**
        $\hat{x}_u \leftarrow f(p)$
        $T \leftarrow \min(t + H, |x|)$     ▷ *Upper time step limit*
        **if** $\text{Bernoulli}(\mu(\text{training\_step})) = 0$ **then**
            $t \leftarrow \arg\min_{t' \in \{t+1..T\}} \mathcal{L}(x_u, x_{t'})$     ▷ *Temporal matching*
        **else**
            $t \sim \text{unif}\{t + 1, T\}$     ▷ *Exploration*
        **end**
        $l \leftarrow l + \mathcal{L}(x_u, x_t)$     ▷ *Accumulate trajectory loss*

        **if** $\text{Bernoulli}(\epsilon(\text{training\_step})) = 0$ **then**
            $p \leftarrow \hat{x}_u$     ▷ *Scheduled sampling (Bengio et al., 2015)*
        **else**
            $p \leftarrow x_t$     ▷ *Take ground truth frame as next model input*
        **end**
        $u \leftarrow u + 1$
    **end**
    Perform a gradient descent step on $\theta$ to reduce $l$
    training_step $\leftarrow$ training_step $+ 1$
**until stopping criterion is reached**;

---

## 6.2 Model architecture

In all experiments, the model $f$ consists of 7 convolutional layers with padding mode "same" and the following specifications, where $\text{Conv}(a, (b, c))$ means "$a$ kernels of size $(b, c)$": $[\text{Conv}(n_k, (5, 5)), \text{Conv}(n_k, (5, 5)), \text{Conv}(n_k, (5, 5)), \text{Conv}(n_k, (7, 7)), \text{Conv}(n_k, (5, 5)), \text{Conv}(n_k, (1, 1)), \text{Conv}(3, (1, 1))]$. Before the 6th layer, the three channels of the model input are concatenated to the feature map. As part of the hyperparameter search, $n_k$ was randomly chosen from the set $\{32, 48\}$. We added two variations of this architecture to the hyperparameter search:

- `f-strided`: the second convolutional layer performs a strided convolution with stride 2 and the 4th convolutional layer performs a transposed convolution.

- `f-dilated`: the fourth convolutional layer uses a dilation rate of 2.

We did not observe substantial difference in the performances of our architectures.

All convolutions were used with a stride of 1. The weight initialization for all layers follows He et al. (2015). We use rectified linear units (ReLU) as activation (Glorot et al., 2011).

## 6.3 Training details

For all experiments, the Adam optimizer (Kingma and Ba, 2014) was used. For hyperparameter search, learning rates for the model $f$ were sampled from the set $\{1.0 \times 10^{-3}, 7.5 \times 10^{-4}, 5.0 \times 10^{-4}\}$.

The learning rate was decayed by a factor of $0.2$ after $n_D$ steps, where $n_D$ was sampled from the set $\{7500, 10000, 15000\}$. The maximum ASI horizon $H$ was sampled from the set $\{15, 18, 21, 25\}$. The number of trajectories per training batch was chosen to be 2 in all experiments. As schedule of exploration for temporal matching we choose $\mu(t) = \max(0, 1 - \frac{t}{K})$, where $K$ was sampled from the set $\{7500, 10000, 15000\}$.

The hyperparameter search described in Section 4 resulted in the parameters shown in Tables 1 and 2, which were used to produce the resulting plots.

| | **ASI** | **ASI w/o exploration** | **fixed** ($\Delta t = 1$) | **fixed** ($\Delta t = 2$) |
|---|---|---|---|---|
| learning rate | $5 \times 10^{-4}$ | $5 \times 10^{-4}$ | $5 \times 10^{-4}$ | $5 \times 10^{-4}$ |
| steps until LR decay | 15000 | 15000 | 15000 | 15000 |
| ASI horizon | 21 | 18 | - | - |
| Exploration steps | 7500 | - | - | - |
| $f$-architecture | f-strided | f-simple | f-simple | f-simple |
| $f$: # of kernels | 48 | 48 | 48 | 48 |

Table 1: Hyperparameters found for Room Runner

| | **ASI** | **ASI w/o exploration** | **fixed** ($\Delta t = 1$) | **fixed** ($\Delta t = 2$) |
|---|---|---|---|---|
| learning rate | $5 \times 10^{-4}$ | $7.5 \times 10^{-4}$ | $7.5 \times 10^{-4}$ | $7.5 \times 10^{-4}$ |
| steps until LR decay | 15000 | 10000 | 15000 | 15000 |
| ASI horizon | 21 | 18 | - | - |
| Exploration steps | 15000 | - | - | - |
| $f$-architecture | f-dilated | f-strided | f-dilated | f-strided |
| $f$: # of kernels | 48 | 32 | 32 | 32 |

Table 2: Hyperparameters found for Funnel Board