[Reviews · NeurIPS 2018]

Reviewer 1



The authors introduced a deep recurrent dynamics model for temporal abstract learning. The proposed method is based on the observation that in many sequential prediction tasks. Throughout this work, there is no any insight technical contribution. Besides, the experimental setting is problematic. There is no evaluate criteria. There is no compared method. The detailed comments are listed as follows, Strength 1) The presentation of this paper is good. 2) the authors give some temporal abstract results. By the qualitative analysis, it shows the effectiveness of the introduced method. Weakness 1) The technical contribution is limited. Throughout this work, there is no any insight technical contribution in terms of either algorithm or framework. 2) The experimental setting is problematic. There is no evaluate criteria. There is no compared method. So we cannot know the superiority of the proposed compared with the existing related methods.

Reviewer 2



## Summary The authors propose a recurrent dynamical model that can jump the prediction time step. To do this, a mechanism called temporal matching, is proposed. In temporal matching, a forward model is encouraged to predict a future time-step in an arbitrary interval (within H) which can be robustly predicted. This is possible because those future time steps with more deterministic expected states provide stronger training signal when trained with the rate-annealed random exploration of the jump time step. To prevent the problem of teacher-forcing, the authors apply scheduled sampling. In Experiments, the quality of the proposed method is evaluated in two tasks, funnel board and room runner, which are expected to perform well for the proposed method. The evaluation is done by applying to a classification task instead of measuring pixel-level reconstruction error. ## Strengths It is dealing with an important problem of jumpy temporal dynamical model which would play a significant role in many applications including model-based RL. The proposed method is relatively simple and thus easy to understand and implement. The idea of matching the future, generated without modelling jump interval, is quite interesting and its interplay with the exploration, which I think is important to avoid optimizing over the argmax in the loss function. ## Weakness The main weakness is experiments. Particularly, the baseline seems very weak because the fixed interval baselines were only for 1 and 2 intervals. To these weak baselines, the superiority of the proposed method seems quite obvious. I think, fixed intervals ranging from 1 to H should be tested and reported. Also, the method does not learn when to finish. Although the authors mentioned that it is not the main focus of the paper, I think it is still important part of a model dealing with this problem. The clarity can also be improved. I recommend to put the Pseudo Code to the main article. Explantation on scheduled sampling is a bit too long although it is not what is proposed in this paper. The paper is written with minimal use of equations and explaining thing more in plain sentences. I think a bit more of formality would help readers understand better.

Reviewer 3



The authors propose a method to learn temporally abstract transition models for time series prediction/classification. Starting with the insight that for many tasks not all time steps need to be accurately predicted, they propose to predict the "best" inputs within a fixed horizon H instead. This is done by evaluating the reconstruction loss for all H future inputs, but back-propagate only on the smallest one, akin to pooling over losses. As a result, the method can learn a sequence of future inputs, with varying delays, that are easy to predict. The target label of a time series can be predicted based either on the last or the average predicted input. The authors evaluate their method on two complex tasks based on input-images to show it outperforms time series prediction with fixed step-sizes. The idea to pool over the loss of future prediction is, to the best knowledge of the reviewer, quite original and appears highly relevant. It is important to distinguish the presented method from model learning for control, though, which have to be conditioned on control signals or actions. The paper is well written and the approach is clearly explained. Experiments are sufficiently complex and show the method's advantage clearly, although the reviewer was missing the comparison with another state-of-the-art algorithm as baseline. Note that the classification problem is somewhat artificially combined with the model-learning, which makes the presented method an unsupervised pre-processing step, rather than a combined classification method. It would be interesting to see if one could backprop the gradient of the classification problem somehow in the model-learning procedure. To this end, the authors are encouraged to try a RNN for classification. RNN yield gradients at each abstract time step, which could be propagated into the model-prediction. The reviewer recommends to accept this paper. It is a well written and presents a simple, but interesting twist on time series prediction. The technique should only work in tasks where the labels depend almost exclusively on the final predicted states, but the basic idea may be applicable beyond that. COMMENTS: l.144: this paragraph was hard to follow l.172+: "separately" twice in one sentence l.221: use the term "NN with seven convolutional layers" instead of "seven-layer fully-convolutional NN" to avoid confusion with seven fully-connected layers l.269: "proposes proposes" fig.7: plotting the best 3 out of 6 runs is a metric that can be easily manipulated and should be avoided fig.9+10: as you mention yourself, these plots are not informative fig.12: Figure 12 refers to itself and the end of the sentence is missing fig.1-12: showing the interquantile range of 3 samples is misleading. Use STD, SEM or minimum/maximum instead